# Effect of Ti_3_SiC_2_ and Ti_3_AlC_2_ Particles on Microstructure and Wear Resistance of Microarc Oxidation Layers on TC4 Alloy

**DOI:** 10.3390/ma15249078

**Published:** 2022-12-19

**Authors:** Gaoyang Gu, Jian Shang, Dan Lin

**Affiliations:** School of Materials Science and Engineering, Liaoning University of Technology, Jinzhou 121001, China

**Keywords:** TC4 alloy, microarc oxidation layer, wear resistance, Ti_3_SiC_2_, Ti_3_AlC_2_

## Abstract

Microarc oxidation (MAO) layers were prepared using 8g/L Na_2_SiO_3_ + 6g/L (NaPO_3_)_6_ + 4g/L Na_2_WO_4_ electrolyte with the addition of 2g/L Ti_3_SiC_2_/Ti_3_AlC_2_ particles under constant-current mode. The roughness, porosity, composition, surface/cross-sectional morphology, and frictional behavior of the prepared MAO layers were characterized by 3D real-color electron microscopy, scanning electron microscopy, X-ray energy spectrometry, X-ray diffractometry, and with a tribo-tester. The results showed that the addition of Ti_3_SiC_2_ and Ti_3_AlC_2_ to the electrolyte reduced the porosity of the prepared layers by 9% compared with that of the MAO layer without added particles. The addition of Ti_3_SiC_2_/Ti_3_AlC_2_ also reduced the friction coefficient and wear rate of the prepared layers by 35% compared with that of the MAO layer without added particles. It was found that the addition of Ti_3_AlC_2_ particles to the electrolyte resulted in the lowest porosity and the lowest wear volume.

## 1. Introduction

Titanium alloys have high specific strength, good corrosion resistance and biocompatibility, a low coefficient of thermal expansion, and are widely used in aerospace, marine, and biomedical materials, but their low hardness and poor wear resistance limit further applications [1,2,3]. The wear and corrosion resistance of titanium alloys can be improved by surface treatment techniques. Current surface techniques for improving titanium alloys include chemical plating, spraying, vapor deposition, laser remelting, microarc oxidation, etc. [4,5,6,7,8]. Microarc oxidation (MAO) is a surface treatment technique used for metals or alloys such as Al [9], Mg [10], and Ti [11]. Under high-voltage or high-current conditions in different electrolytes, the ceramic layers are grown in situ on the material surface under the action of thermochemistry, plasma chemistry, and electrochemistry [12,13]. The current research on microarc oxidation is mainly focused on adjusting the electrolyte, optimizing the electrical parameters, and post-treatment sealing holes to improve the material properties. In comparing the different electrolyte systems, researchers found that the MAO layer in the silicate electrolyte grew almost entirely outwards, which resulted in a porous coating and improved wear resistance. The inward growth of the MAO layer in the phosphate electrolyte resulted in a compact coating structure and high adhesion, but the wear resistance of the coating was poor [14]. At the same time, researchers have added additives such as sodium tungstate, sodium molybdate, CNTs, and rare earth salts to the electrolyte to improve the density of the film layer and have also changed electrical parameters such as the power mode, current density, frequency, and duty cycle to control the thickness and roughness of the MAO layer [15,16,17,18,19,20,21,22].

In recent years, many researchers have added insoluble nanoparticles of SiO_2_, ZrO_2_, SiC, TiO_2_, and Al_2_O_3_; graphite powder; graphene; and other compounds to the electrolyte to prepare microarc oxidation layers to improve wear resistance, thickness, and roughness, and reduce defects [23,24,25,26,27,28,29]. Ti_3_SiC_2_ and Ti_3_AlC_2_ have high melting points and good corrosion resistance as well as the high thermal and electrical conductivity of metals [30]. There are no studies on the addition of titanium–silicon–carbon to the electrolyte. In this study, we prepared microarc oxidation layers by adding Ti_3_SiC_2_/Ti3AlC_2_ conductive particles to the electrolyte and compared and analyzed the structure, composition, and wear resistance of the prepared layers.

## 2. Experiments and Characterization

### 2.1. Preparation and Experiments

The TC4 alloy rods were used to make specimens of size Φ25 × 3mm, which were then polished with 120#~800# sandpaper, cleaned with anhydrous ethanol ultrasonically, and blow-dried. The electrolyte was composed of a compound salt system (8g/L Na_2_SiO_3_ + 6g/L 6g/L (NaPO_3_)_6_ + 4g/L Na_3_WO_4_), and 2g/L Ti_3_SiC_2_/Ti_3_AlC_2_ was added to the electrolyte to make a suspension. The microarc oxidation process was conducted with a WHD-30D plasma microarc oxidation system with electrical parameters of a constant current of 11A/dm^2^, duty cycle of 50%, frequency of 500 H_Z_, and time of 15 min. Table 1 shows the information on the main experimental materials and reagents.

### 2.2. Analysis and Characterization

D/Max-2500 X-ray diffractometer was used to analyze the physical phase composition of the prepared layer (20°~90°, 6°/min); Axio Vert.A1 metallographic microscope and Zeiss SIGMA 500 scanning electron microscope were used to observe the cross section and surface morphology of the layer; a pin–disk friction and wear tester was used to conduct a dry sliding test on the MAO layer. SEM photographs of the surface of the MAO layer were analyzed using Image J software, and the original image threshold was adjusted to cover the pores. The percentage of area occupied by surface pores was calculated as the porosity of the oxide film using a software calculation tool [31,32]. Film thickness and wear volume were measured several times and averaged to ensure quality control.

## 3. Results and Discussion

### 3.1. Cross Section and Surface Topography

From Figure 1, it can be seen that the thickness of the MAO layer without the addition of particles is about 17~25 μm. The inner layer is dense, and the surface layer is loose. There are small micropores in its “honeycomb” structure and a few cracks in some areas. By adding the particles to the electrolytes, the thickness of the layer slightly increases, there are no large pores, and the density of the MAO layer is higher. As shown in Figure 2, the porosity of the MAO layer was obviously reduced after adding Ti_3_SiC_2_/Ti_3_AlC_2_. As shown in Figure 3, the thickness of the MAO layer after adding Ti_3_SiC_2_/Ti_3_AlC_2_ was less affected by the added particles, increasing by 1–2 μm, while the porosity decreased by about 46% after adding Ti_3_AlC_2_ particles compared with MAO. The reason for these two phenomena is that the addition of the MAO discharge stage produces a violent spark discharge temperature, which penetrates the MAO layer formed in the low-pressure stage and forms a discharge channel, from which molten metal oxides are ejected onto the surface of the MAO layer, and defects such as surface micropores and pits are formed through quenching, cooling, and curing of the electrolyte. Cracks are formed when the surface temperature is too high, and stress cracks appear during solidification when the subcooling degree is high. On one hand, the addition of Ti_3_SiC_2_/Ti_3_AlC_2_ to the electrolyte improves the conductivity and reduces the arc starting voltage of MAO. On the other hand, the high melting point of ceramic particles absorbs the heat generated by the discharge spark and produces a small amount of microporosity on the uniform surface [33,34].

### 3.2. X-ray Analysis

As can be seen from Figure 4, the MAO layer is mainly composed of anatase and rutile phases and a small amount of matrix and second-phase particles. It is shown that the high MAO surface temperature causes the molten oxidation of the matrix surface, and the TiO_2_ transforms from an indeterminate form to a substable anatase phase due to the slow atomic diffusion caused by the high subcooling, while the matrix surface temperature is higher and the substable crystallization forms the rutile phase [35]. The very high diffraction peaks of the TC4 matrix appear in the XRD pattern due to the loose structure of the surface layer, which is easier for the rays to penetrate, and the dense and more stable structure of the inner layer. The addition of Ti_3_SiC_2_/Ti_3_AlC_2_ did not change the phase composition of the MAO layer.

### 3.3. Friction and Wear Experiment

From Figure 5a, it can be seen that the friction coefficient gradually increases from low to high with the increase in friction time. This is due to the sparse surface structure of the MAO layer, its roughness, and its low hardness. The friction coefficient is low at the beginning of the friction; with the increase in friction time, the particles fill the micropores and then the actual contact surface of the dense structure. The friction coefficient gradually increased, and then tended to stabilize and then fluctuate. The friction coefficient curve after the addition of the particles all decreased. From Figure 5b, it can be seen that the average friction coefficient decreased with Ti_3_SiC_2_/Ti_3_AlC_2_ particle addition, and wear volume decreased from 0.92 mm^3^ to 0.59 mm^3^. From Figure 6, it can be seen that after sliding friction, the abrasion marks obtained are deep and there are a lot of deep grooves and protrusions on the rough surface of the abrasion marks (Figure 5a), while after the addition of Ti_3_SiC_2_/Ti_3_AlC_2_ particles, the abrasion marks of the layer are smooth and shallow. The addition of Ti_3_SiC_2_/Ti_3_AlC_2_ particles to the prepared layer can reduce wear volume. This wear reduction is because of the Ti_3_SiC_2_/Ti_3_AlC_2_ particles in the MAO layer that provide self-lubrication during the frictional wear process [36].

Figure 7a shows that there are many abrasive particles and large grains in the abrasion marks of the MAO layer without added particles. Transverse cracks appear in the grooves and the wear mechanism is abrasive wear. From Figure 7b, it can be seen that the surface of the abrasion marks of the MAO layer with Ti_3_SiC_2_ particles appears to be delaminated. This is due to the large difference in hardness between the dense layer and the substrate and the stress concentration of the flaky abrasive particles during wear. The wear surface of the MAO layer containing Ti_3_AlC_2_ ceramic particles is smooth, and the wear depth is shallow, as seen in Figure 7c. The improved wear marks are attributed to the lubricity properties of the particles and the improved quality of the MAO layer. An EDS surface scan of the wear marks was carried out. The elemental composition of the wear marks changed significantly after the particles were found. As shown in Figure 8a, only a large amount of Ti and O and a small amount of matrix elements such as Al and V are present in the wear marks of the MAO layer. In (b,c), a certain amount of Si appears, and the content of Al increases in (c). This can be seen in the energy spectrum dot scan at the yellow circle mark in Figure 9; in the MAO grinding without added particles, the particles in the grinding marks are 90% Ti and O. This means that the main component is TiO_2_. However, the addition of Ti_3_SiC_2_ in Figure 9b results in a significant increase in Si and C, indicating the presence of Ti_3_SiC_2_, with SiO_2_ in the particles. The increase in Al and C in Figure 9c indicates the presence of Ti_3_AlC_2_ as well as Al_2_O_3_. This is also consistent with the findings of the surface scan in Figure 6. The presence of small amounts of Ti_3_SiC_2_/Ti_3_AlC_2_ on the wear marks can be attributed to two factors: the original particles involved in the growth of the wear marks, and the shallow particles carried by the grinding ball as it was worn during the grinding process. However, the abrasion marks penetrate deep into the matrix and only a few Ti_3_SiC_2_/Ti_3_AlC_2_ particles are attached to the abrasion marks. This suggests that the Ti_3_SiC_2_/Ti_3_AlC_2_ present in the MAO layer is involved in the wear process and does have a positive effect on the wear resistance.

### 3.4. Wear Mechanisms

The effect on the layer thickness and porosity is mainly due to two factors: Firstly, Ti_3_SiC_2_/Ti_3_AlC_2_ are deposited directly on the layer and have a “sealing effect”. Secondly, Ti_3_SiC_2_/Ti_3_AlC_2_ increase the electrical conductivity of the electrolyte, which affects the voltage during MAO. As can be seen in Figure 9, no significant difference appears in the first 30s of MAO, and after 30s the voltage of MAO without added particles is significantly higher than the other two layers. At this point, the low voltage means low energy density, which also leads to low damage to the film layer and ultimately to a significant increase in MAO layer density. In Figure 10b, it can be seen that during the first 200S of the microarc oxidation process there is a certain difference in current: the addition of Ti_3_SiC_2_/Ti_3_AlC_2_ causes an extremely higher current curve, corresponding to a lower voltage at this stage than without the addition of these particles, which again proves that Ti_3_SiC_2_/Ti_3_AlC_2_ increase the electrical conductivity of the solution. In the subsequent stages, the addition of pellets did not have a significant effect on the current profile of the microarc oxidation process due to the constant-current mode.

The effect of Ti_3_SiC_2_/Ti_3_AlC_2_ on wear resistance is divided into three stages, as shown in Figure 11: (a) The first stage, when the sparse surface oxide layer is in contact with the abrasive ball and the change in the surface quality of the film does not significantly improve the average coefficient of friction. (b) The second stage, when the sparse surface layer of microarc oxidation has been worn through. From the first to the second stage, the pores of the sparse oxide layer and the Ti_3_SiC_2_/Ti_3_AlC_2_ in the film gradually start to influence the friction process. (c) The third stage, when the MAO layer is worn through, and a certain amount of the titanium alloy matrix is exposed. Ti_3_SiC_2_/Ti_3_AlC_2_ play two main roles in the process from the second to the third stage. As particles in a laminar structure, they have good self-lubricating properties, so a certain amount of Ti_3_SiC_2_/Ti_3_AlC_2_ is concentrated to form a lubricating layer under the high-temperature and high-pressure conditions of the grinding ball friction process. The friction coefficient of the film layer is reduced. On the other hand, the added particles decompose and oxidize during the friction process, forming hard phases such as SiO_2_ and Al_2_O_3_, which increase the hardness of the MAO layer. It can be concluded that the first to second stage corresponds to the period before 4 min in Figure 5, when the Ti_3_SiC_2_/Ti_3_AlC_2_ particles have not yet had sufficient influence so that there is no significant difference in the friction coefficient curve, and the second to third stage corresponds to the period after 4 min in Figure 5, when the friction coefficient curve under the Ti_3_SiC_2_/Ti_3_AlC_2_ particles slowly decreases and fluctuates under the influence of the particles. The friction coefficient of the MAO layer with the added particles at this stage is low, wear is low, and the wear resistance of the MAO layer is improved.

In conclusion, the addition of Ti_3_SiC_2_/Ti_3_AlC_2_ have a positive effect on the thickness, porosity, and wear resistance of the MAO layer.

## 4. Conclusions

The MAO layers prepared by adding Ti_3_SiC_2_ and Ti_3_AlC_2_ particles to the electrolyte are thicker than those without added particles. The thickest MAO layer with the addition of Ti_3_SiC_2_ particles is 21.42 μm, and the lowest surface porosity with the addition of Ti_3_AlC_2_ is 46% lower that of the MAO layer without added particles. The addition of Ti_3_AlC_2_ particles to the electrolyte reduces the coefficient of friction and improves wear resistance. The layer prepared by adding Ti_3_AlC_2_ particles to the electrolyte has an average friction coefficient of 0.37 and a wear volume of 0.5939 mm^3^, which is 35% less than that of the MAO layer without added particles. It is clear that the addition of Ti_3_AlC_2_ to the electrolyte has a positive effect on the thickness, porosity, and wear resistance of the MAO layer.

## Figures and Tables

**Figure 1 materials-15-09078-f001:**
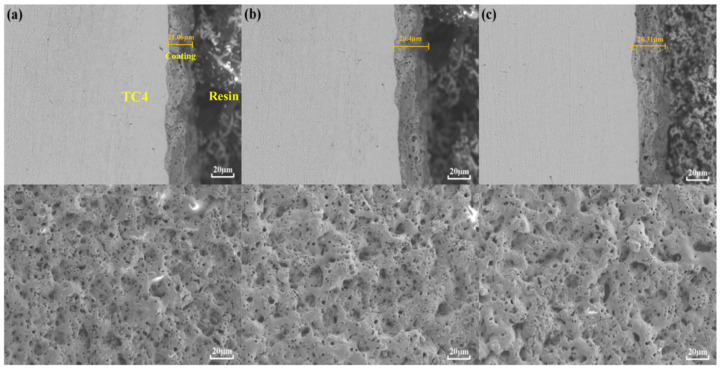
Cross-sectional and surface morphology of MAO layer on TC4 alloy with different additives: (**a**) MAO, (**b**) Ti_3_SiC_2_, (**c**) Ti_3_AlC_2_.

**Figure 2 materials-15-09078-f002:**
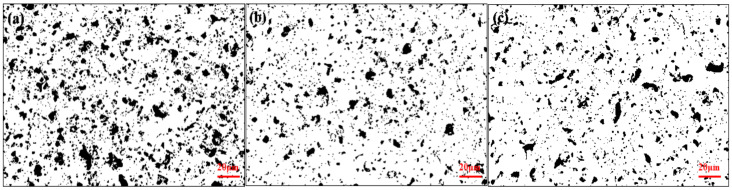
Surface porosity of MAO layers on TC4 alloy with different additives (pores marked in black): (**a**) MAO, (**b**) Ti_3_SiC_2_, (**c**) Ti_3_AlC_2_.

**Figure 3 materials-15-09078-f003:**
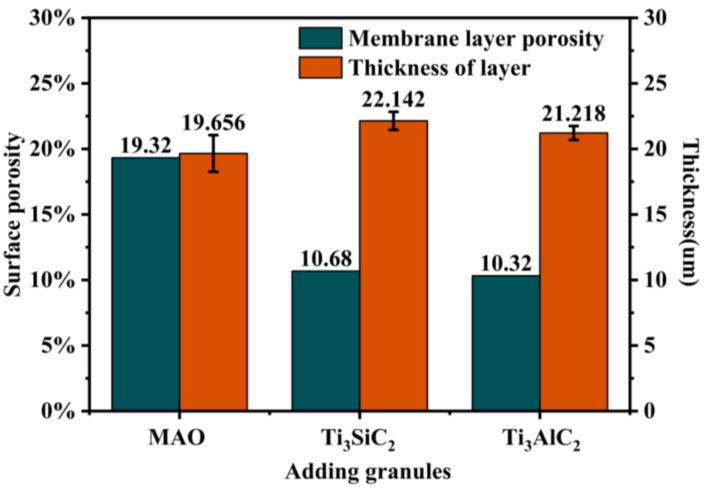
Surface porosity and average MAO layer thickness on TC4 alloys with different particle additions.

**Figure 4 materials-15-09078-f004:**
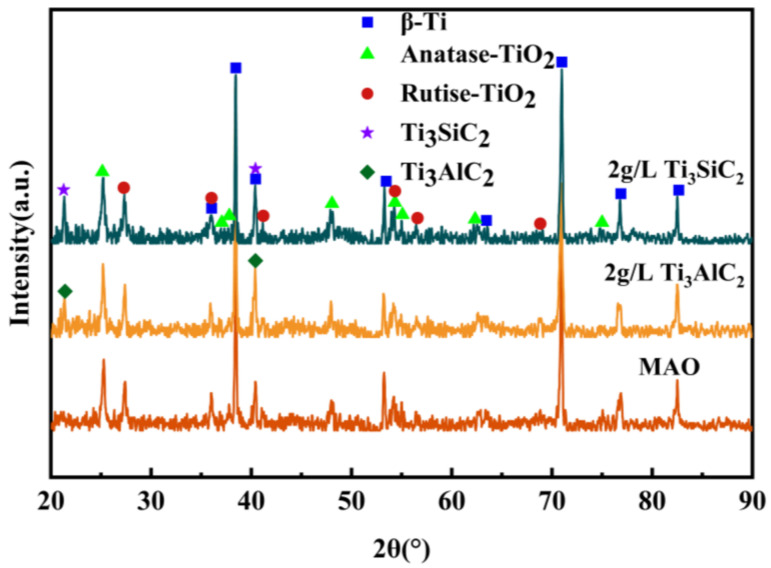
X-ray diffraction patterns of MAO layer on TC4 alloy with different particle additions.

**Figure 5 materials-15-09078-f005:**
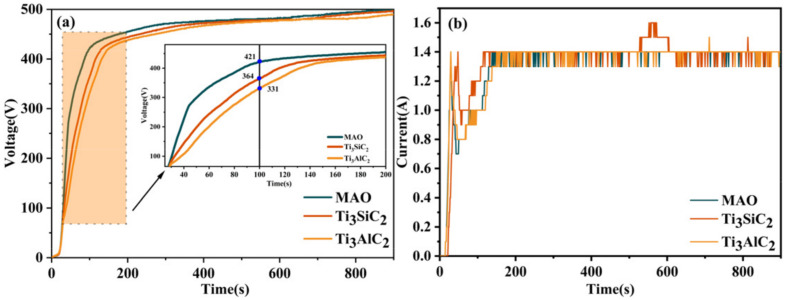
Results of (**a**) dynamic friction coefficient and (**b**) average friction coefficient and wear volume loss of MAO layer onTC4 alloy with different particle additions.

**Figure 6 materials-15-09078-f006:**
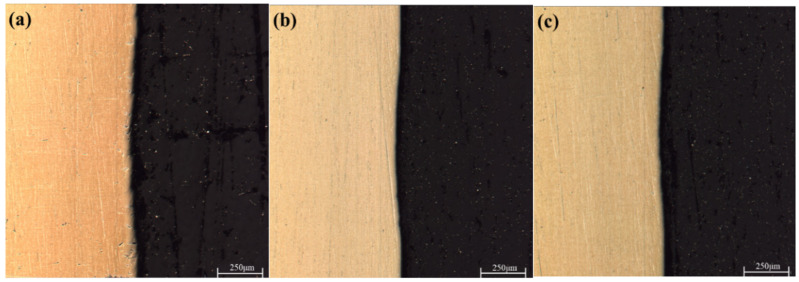
Cross-sectional morphology of MAO layers on TC4 alloys with different particle additions. (**a**) MAO, (**b**) Ti_3_SiC_2_, (**c**) Ti_3_AlC_2_.

**Figure 7 materials-15-09078-f007:**
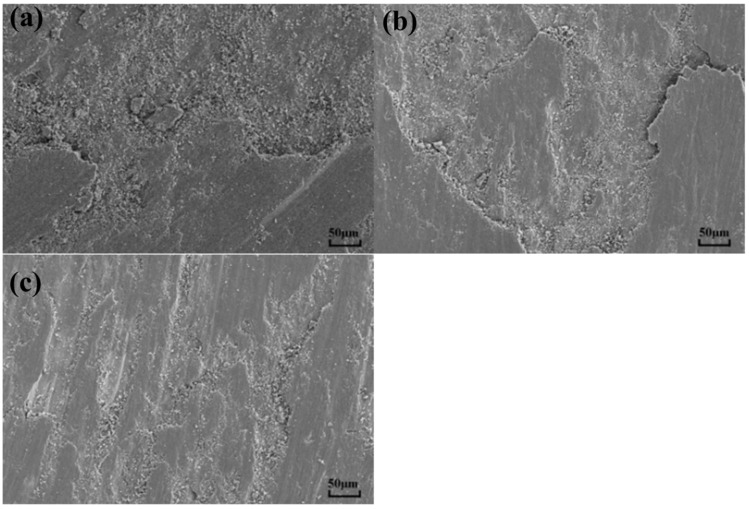
Worn morphology of MAO layer on TC4 alloy with different particle additions: (**a**) MAO, (**b**) Ti_3_SiC_2_, (**c**) Ti_3_AlC_2_.

**Figure 8 materials-15-09078-f008:**
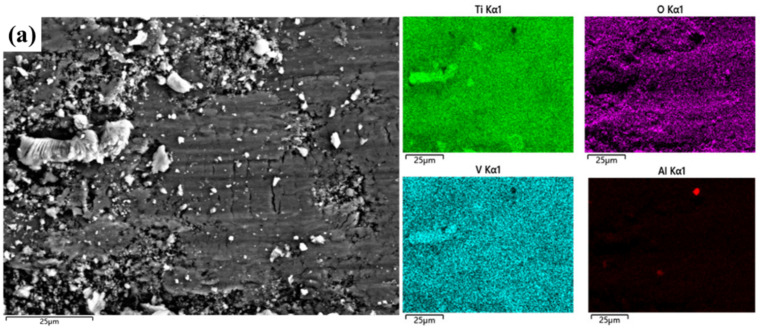
EDS of MAO layer on TC4 alloy with different particle additions: (**a**) MAO, (**b**) Ti_3_SiC_2_, (**c**) Ti_3_AlC_2_.

**Figure 9 materials-15-09078-f009:**
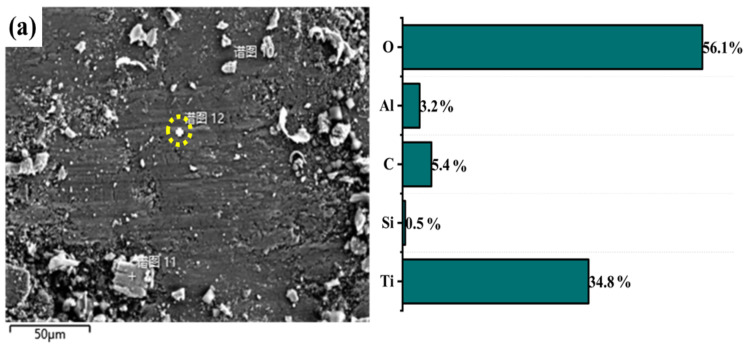
EDS results of worn surface with different particles: (**a**) MAO, (**b**) Ti_2_SiC_3_, (**c**) Ti_2_AlC_3_.

**Figure 10 materials-15-09078-f010:**
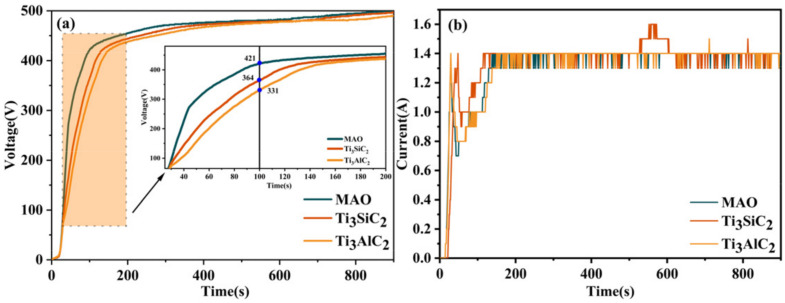
Voltage vs. current curves for MAO layers on TC4 alloys with different particle additions. (**a**) Voltage curve. (**b**) Current curve.

**Figure 11 materials-15-09078-f011:**
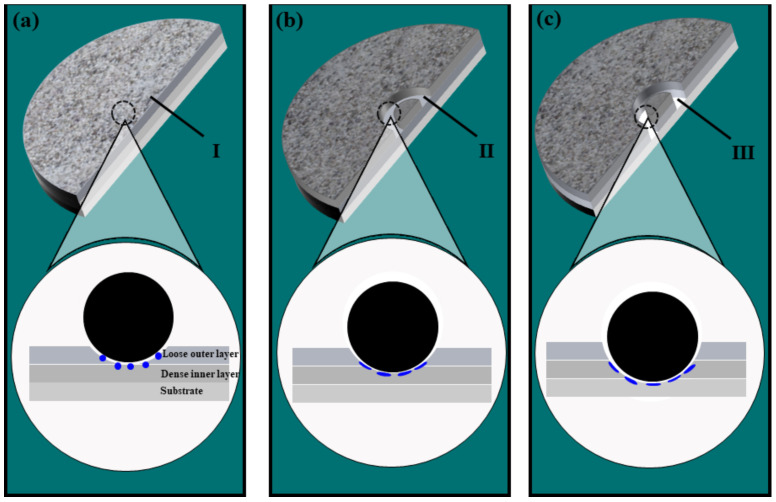
Wear mechanism diagram of prepared layer with Ti_3_SiC_2_ and Ti_3_AlC_2_ particles addition. (**a**) Friction Stage 1. (**b**) Friction Stage 2. (**c**) Friction Stage 3. I Loose outer layer. II Dense inner layer. III Substrate.

**Table 1 materials-15-09078-t001:** Experimental reagents and materials.

Name	Formula	Purity Manufacturer	Country
TC4	-	HuiJing Metal Materials Limited	China
Na_2_SiO_3_	AR	Sinopharm Chemical Reagent Co.	China
Na_3_PO_4_	AR	Sinopharm Chemical Reagent Co.	China
Na_2_WO_4_	AR	Sinopharm Chemical Reagent Co.	China
Ti_3_SiC_2_	98%	Forsman	China
Ti_3_AlC_2_	98%	Forsman	China

## Data Availability

Not applicable.

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
