# Peer review of "Effect of Ti3SiC2 and Ti3AlC2 Particles on Microstructure and Wear Resistance of Microarc Oxidation Layers on TC4 Alloy"

_materials, 2022, doi:10.3390/ma15249078_

Round 1
Reviewer 1 Report
The research article ‘Effect of Ti3SiC2 and Ti3AlC2 particles on microstructure and wear resistance of micro-arc oxidation layers on TC4 alloy’
In this research, the miro acr oxidation of TC4 alloy specimens by using 8g/L Na2SiO3+6g/L (NaPO3)6+4g/L Na2WO4 electrolyte with the addition of 2g/L Ti3SiC2/Ti3AlC2 particles under constant current mode. Characterization of the samples was made by focusing on microstructure and wear resistance. There are a few remarks related to this work:
· All images are too small and too low quality. It is impossible anything to see/read in them.
· Fig. 2 and in general the method for evaluation of porosity is questionable. This method shouldn’t be applied to the evaluation of porosity, BET analysis is a golden standard for the evaluation of porosity.
· In Fig. 3. How porosity was calculated? Fig. 2. SEM images are not enough to state about such a high porosity as it is clearly not true. BET analysis is required.
· Section 2.3. shouldn’t start from the image.
· In all cross-section images a scale to show the layer thickness required.
· Fig 6-7 can’t discuss anything about as images are too small and information in them is invisible.
· Conclusions should be rewritten and more specific with actual numbers/parameters/size etc.
At the moment, the article requires improvements.
Reviewer 2 Report
1. In paragraph 1, it is necessary to add manufacturing companies and countries for each material, chemical reagent, and equipment used.
2. In paragraph 2, it is necessary to add graphs of currents and voltages of coating formation and also reflect in the discussion the influence of various additives on the stages of coating formation.
3. Figure 1 It is necessary to add the designation of the base metal, coating, and polymer to the cross-section picture, to distinguish them.
4. In paragraph 1.2, Please explain how surface porosity was measured.
5. ‘’The addition of Ti3SiC2/Ti3AlC2 to the electrolyte improves the conductivity of the electrolyte on the one hand and reduces the arc starting voltage of MAO. On the other hand, the high melting point of ceramic particles absorbs the heat generated by the discharge spark and produces a small amount of microporosity on the uniform surface ‘’. This statement must be proved or referred to by other researchers.
6. Line 98 Why do the introduced particles of carbides under the influence of high temperature in the microplasma channel do not burn out?
7. Figure 4 It is necessary to add (a) and (b) in the figures themselves.
8. Corrosion studies are written for the purpose of the work, but they are not in work.
In this article, a very narrow range of characterization tools is used, and the explanation of the results obtained is superficial without clarifying the effect of the introduced additives.
Round 2
Reviewer 1 Report
The research article 'Effect of Ti3SiC2 and Ti3AlC2 particles on microstructure and wear resistance of micro-arc oxidation layers on TC4 alloy'
In this research, the miro acr oxidation of TC4 alloy specimens by using 8g/L Na2SiO3+6g/L (NaPO3)6+4g/L Na2WO4 electrolyte with the addition of 2g/L Ti3SiC2/Ti3AlC2 particles under constant current mode. Furthermore, the samples were characterized by focusing on microstructure and wear resistance.
The Authors made some improvements and updated the manuscript. However, there is some remark related to this work:
· Fig. 2. isn't apparent. What marks white/black colors? It should be explained in the caption.
· Fig. 3. why did your calculated results of porosity decrease from the last reviewed manuscript version? It is still not clear how the porosity is calculated. Also, there is no information about these calculations' quality control (QC). Error bars should be added in fig. 3.
· Do the Authors of this work calculate the surface that pores occupy on the surface of the samples? In this case, it should not be stated as porosity.
· What about quality control (QC) in other measurements? It should be explained in the experiments and characterization section. Error bars should be added in Fig. 4.
· Usually, conclusions are written as a short and consistent summary paragraph of the results. For the Authors, it would be reasonable to consider rewriting conclusions in this way.
Reviewer 2 Report
Dear authors,
Your manuscript looks much better, but you have not removed the study of corrosion properties from the objectives of the work (Lines 47-48).
Preferred citation of the journal Materials.
In addition, the references in the introduction need to be updated because the references from 2005 - 2012 are pretty old.
Also, line 26 must be added regards to enumerated metals; I propose adding the following:Al:
Aljohani, T.A.; Alawad, M.O.; Elkatatny, S.; Alateyah, A.I.; Rubayan, M.T.B.; Alhajji, M.A.; AlBeladi, M.I.; Khoshnaw, F.; El-Garaihy, W.H. Electrochemical Behavior of SiC-Coated AA2014 Alloy through Plasma Electrolytic Oxidation. Materials 2022, 15, 3724. https://doi.org/10.3390/ma15103724
Mg:
Bisztyga-Szklarz, M.; Rząd, E.; Boroń, Ł.; Klimczyk, P.; Polczyk, T.; Łętocha, A.; Rajska, M.; Hebda, M.; Długosz, P. Properties of Microplasma Coating on AZ91 Magnesium Alloy Prepared from Electrolyte with and without the Borax Addition. Materials 2022, 15, 1354. https://doi.org/10.3390/ma15041354
Ti:
Schwartz, A.; Kossenko, A.; Zinigrad, M.; Gofer, Y.; Borodianskiy, K.; Sobolev, A. Hydroxyapatite Coating on Ti-6Al-7Nb Alloy by Plasma Electrolytic Oxidation in Salt-Based Electrolyte. Materials 2022, 15, 7374. https://doi.org/10.3390/ma15207374
